# Keeping Morality “on the Straight” and Never “on the Bend”: Metaphorical Representations of Moral Concepts in Straightness and Curvature

**DOI:** 10.3390/bs13040295

**Published:** 2023-03-30

**Authors:** Xiaoyan Zhu, Yanbing Huang, Wenxuan Liu, Zhao Yu, Yan Duan, Xianyou He, Wei Zhang

**Affiliations:** 1Center for Studies of Psychological Application, Key Laboratory of Brain, Cognition and Education Sciences, South China Normal University, Guangzhou 510631, China; 2Ministry of Education, Guangdong Key Laboratory of Mental Health and Cognitive Science, School of Psychology, South China Normal University, Guangzhou 510631, China; 3Key Laboratory of Chinese Learning and International Promotion, South China Normal University, Guangzhou 510631, China

**Keywords:** moral concept, curve and straight metaphorical representation, implicit association test, Stroop paradigm, embodied cognition

## Abstract

The study of moral conceptual metaphors has been an important topic in recent years. In Chinese culture, the concepts of curvature and straightness are given certain semantic contents, in which curvature refers to being sly while straightness refers to having integrity. In the present study, we used the Implicit Association Test (IAT) paradigm (Experiment 1) and the Stroop paradigm (Experiment 2) to investigate whether there are metaphorical representations of curvature and straightness in moral concepts. The results revealed that the mean reaction time in compatible trials (i.e., moral words accompanied by a straight pattern and immoral words accompanied by a curved pattern) was significantly shorter than that in incompatible trials (i.e., moral words accompanied by a curved pattern and immoral words accompanied by a straight pattern). The Stroop paradigm showed that reaction times were significantly reduced when moral words were presented in a straight font, but there was no significant difference between the presentation of immoral words in a straight font and that in a curved font. The results suggest that mental representations of moral concepts are associated with straightness and curvature in Chinese culture.

## 1. Introduction

A metaphor is the mapping of concrete concepts to abstract concepts; that is, it is a way to express and understand abstract concepts with the help of concrete concepts. People use metaphors more commonly as a process of representing unfamiliar abstract concepts based on life experience [1]. In other words, metaphor comprehension is based on extensions of the same processes that underlie thinking and language comprehension in general [2]. For a long time, people have regarded metaphors as rhetorical devices and have not considered the connection between metaphors and reality, let alone the influence of metaphors on the cognitive process. As a kind of cognition, a metaphor interacts with the situational context in various ways. On the one hand, metaphor processing relies on conceptual representation, and conceptual representation includes both linguistic composition and embodied composition, which means that a metaphor is an interactive process that adapts to a particular situation [3]. The activation levels for different metaphorical representations vary according to their contextual situation, and people will activate the form of metaphorical representation that best fits their current situation [4]. On the other hand, if placed in the proper context, metaphorical representations help to increase the speed of comprehension, thereby increasing ‘processing fluency’ and enhancing preferences [5]. For example, some studies have found through the Stroop paradigm and the IAT that in the Chinese cultural context, response time is significantly shorter when high power is presented with the color gold, and low power is presented with the color grey, indicating that people tend to use gold to represent high power and grey to represent low power; this phenomenon may be closely related to Chinese language, culture and life [6].

In conceptual metaphor theory, it is claimed that mental processes include the imitation of body-related perceptions and actions and that people’s retrieval of conceptual meaning involves the partial representation of sensory and motor experiences [7]. This effect is particularly evident in the processing of moral concepts because the highly abstract nature of moral concepts relies on the use of representational metaphors. In recent years, empirical studies on moral metaphors have also focused on the mutual representational relationships between moral concepts and physical attributes, such as the sensory dimensions of cleanliness, color, lighting, and space. Research regarding the moral cleanliness metaphor points to the common belief among many cultures, especially religious cultures, that “cleanliness is virtue”, as in the case of Buddhists who bathe and purify themselves before burning incense to worship Buddha.

In daily life, people often use linguistic metaphors to express the moral concepts in straightness and curvature. For example, “courage is the ladder on which all the other virtues mount” and “by hook or by crook”. In Chinese, people also use words such as “清身洁己” (qīng shēn jíe jǐ, which refers to maintaining one’s own moral integrity and physical conduct) and “冰清玉洁” (bīng qīng yù jíe, which refers to noble and pure personality traits) to describe people who behave morally and words such as “脏心烂肺” (zāng xīn làn fèi, which refers to people who are sordid and devious) and “脏官污吏” (zāng gūan wū lì, which refers to corrupt officials) to describe people who behave immorally. Some studies have also found that participants rated the moral dimension of social issues lower after having washed their hands or having read an article on physical cleanliness, suggesting that activating participants’ self-cleaning impulse can lead to their making more stringent moral judgements [8]. Through field and laboratory experiments, researchers found that participants who had handled and counted dirty money tended to behave selfishly and unfairly, while those who had counted clean money tended to behave reciprocally and fairly [9]. Similarly, one can also obtain psychological compensation from cleaning behavior after engaging in unethical behavior. For example, after engaging in unethical behaviors such as lying, people would choose to use mouthwash or hand sanitizer to clean the ‘soiled’ area [10]. A previous study also applied eye-tracking technology in order to track pupil size before and after hand washing as an objective measurement for subjects’ emotional arousal, and they found that compared to the hand-washing group, the no-washing group had extreme rating scores for morality. Meanwhile, the eye-tracking results also showed a data pattern of decreased pupil diameter after hand washing [11]. In separated Stroop paradigm and event-related potentials (ERPs) experiments, participants recognized moral words faster after reading environmentally clean sentences than neutral sentences, and the amplitude was more positive in the P300 and N400 time windows. Meanwhile, they also estimated moral words faster after reading self-cleaning sentences than neutral sentences, and the amplitude was more positive in the P300 and N400 time windows [12,13]. In functional magnetic resonance imaging experiments, participants engaged in unethical behavior were more satisfied with cleaning products than those engaged in ethical behavior, and their cortical networks involving sensorimotor brain regions were significantly activated during this period, demonstrating that the understanding of ethics can be grounded in sensory experience [14]. Although the act of self-cleansing can reduce the guilt and regret that individuals feel after performing an immoral act, different forms of cleansing do not have the same impact in Eastern and Western cultures. For example, the moral benefits of hand washing are more pronounced among Westerners than East Asians, while the moral effects of facial cleaning are more powerful among East Asians than Westerners [15]. Moreover, the concept of cleanliness includes not only physical cleanliness but also environmental cleanliness. A researcher conducted a field experiment with corporate employees and found that environmental cleanliness significantly influenced employees’ judgments of unethical work behaviors (e.g., organizational retaliation, transgressions, and counterproductive behaviors), while the effect of self-cleanliness was nonsignificant, which is not consistent with the results of previous studies. The researcher explained that this inconsistency may be because the moral judgement of self-cleanliness mainly involves the personal moral domain, while environmental cleanliness focuses on the organizational moral domain, and the moral metaphors of the two may have different mechanisms [16].

In terms of moral color metaphors, some studies have found that there is a metaphorical relationship between the concept of morality and the spectrum of black and white, and that this relationship has cross-cultural consistency. White represents the positive and beautiful side of things, and black represents the dark side. For example, in the Stroop task, people named colors significantly faster when immoral words were presented in black compared to moral words being presented in black, and conversely, people named colors significantly faster when moral words were presented in white compared to immoral words being presented in white [17]. In addition, it was also found that when comparing the use of a greyscale background to that of a blue and yellow checked background, participants gave ratings that were significantly further from the midpoint of the scale when making judgements about Heinz’s moral dilemma of drug theft and a range of other moral issues presented in front of a black and white checked background. That is, their views become more extreme, suggesting that incidental visual experiences related to black and white color contrasts can influence people’s moral judgements and lead them to make more extreme moral judgements [18].

Another metaphorical relationship similar to the moral color metaphor is the moral lucidity metaphor [19,20]. Chinese people often use expressions such as “弃暗投明” (qì àn tóu míng, which refers to forsaking darkness for light) and “明人不做暗事” (míng rén bú zùo àn shì, which refers to an honest man does not do anything underhand), implying that “brightness” contains a positive meaning and “dark” contains a derogatory meaning. On the one hand, studies have found that people who play the Dictator game in a bright environment show more prosocial behavior than those who play it in a dark or only moderately bright environment [21]. On the other hand, moral processing can also affect people’s own perceptions of the brightness of their surroundings. In one study, participants were asked to recall moral or immoral events that they had experienced in the past and describe them emotionally, and it was found that recalling immoral events caused participants to perceive their environment as darker than they otherwise would and to show a preference for brighter objects [22]. A meta-analysis also found that recalling past moral behavior can influence individuals’ perceptions of luminosity. This may be because people gradually develop the experience of avoiding the dangers of dark environments over time, and thus gradually derive a metaphorical relationship between morality and luminosity [23].

In addition to moral color metaphors, the empirical study of moral metaphors also includes the study of moral spatial metaphors. Spatial metaphors refer to the way people often use familiar spatial concepts such as up and down, left and right, inside and outside, and far and near to represent certain abstract concepts [24], such as mapping concrete spatial relations to moral concepts. A researcher using the IAT asked participants to discriminate based on the nature of the presented words, and the results showed that the correct response rate was significantly higher and the response time significantly shorter when moral words were presented at the top of the screen and immoral words were presented at the bottom [25]. In another study, the cognitive mechanisms underlying moral metaphors on the vertical dimension was explored, and the results revealed a stronger association of “moral/up, immoral/down” when the emotionality of moral and immoral words was higher than when it was lower [26]. The event-related potential technique can also be used to demonstrate a metaphorical relationship between morality and vertical space, which occurs early in target recognition, as the “moral/up” association modulates the amplitude of the N1, P2, and late positive-going potential when processing moral words, whereas the “immoral/down” association only modulates the amplitude of the late positive-going potential when processing immoral words [27].

In addition to the vertical dimension, the horizontal dimension is rich in metaphors. Researchers have found in the USA that statues of heroes such as Mother Teresa, Martin Luther King, Jr. and Nelson Mandela gaze to the right of the viewer because in the Western world, right is preferred to left; right is associated with personal virtues such as interpersonal warmth, pride and a sense of the future, and the rightwards looking pose conveys a greater level or heroism [28]. However, distinct cultural contexts use different orientation metaphors. There are differences between the metaphorical association of orientation with moral concept levels in the West and that in the Chinese context. Researchers in Chinese culture have demonstrated the existence of the metaphorical mapping of morality to the left and immorality to the right from different perspectives [29]. In addition, there is also a metaphorical relationship between moral concepts and being on the inside or outside of a conceptual container. One study found that people responded more quickly when moral words appeared inside the container than when they appeared outside the container, while the response time was shorter when immoral words appeared outside the container. This may be because the outside of the container symbolizes transgression and is therefore easily associated with the concept of immorality on a metaphorical level, while the inside of the container symbolizes normality and is therefore easily associated with the concept of morality [30].

Additionally, there are other areas of moral metaphor research. Some researchers have suggested that transparency is a paradoxical moral metaphor, which contains both a positive concept containing ideas of “social responsibility, social justice, environmental safety, and true democracy” and a negative concept associated with “invisibility, pretense, lawlessness, and violence” [31]. In addition, some researchers have studied the olfactory sensory dimension, indicating that fresh and clean smells can promote benign behaviors such as volunteering and donating and can increase tendencies to reciprocate trust and provide charitable help [32]. Electromyography-based techniques also found a high degree of similarity in the facial motor activity elicited by people experiencing taste aversion (caused by unpleasant tastes), basic aversion (caused by photographs of pollutants), and moral aversion (caused by unfair treatment in economic games), and researchers found that bad tastes elicited the same nasal and oral rejection responses as moral aversion did, suggesting that bad tastes may be associated with immorality through metaphorical linkage [33].

The bidirectional representational relationship between moral concepts and physical attributes shows that impurities causing core disgust, such as dirt and stink, can affect behavior and spatial dimensions of moral judgment. For example, research showed that both moral violations involving and not involving impurity promoted the detection of disgusted faces [34], and moral violations with impurity cues elicited a higher severity of moral judgments than those without impurity cues [35]. However, this relationship was modulated by the degree of core disgust priming; one study had found that moral severity ratings for moral transgressions after the presentation of neutral images were significantly higher than those after the presentation of highly disgusting images [36].

In addition, we often use the concepts of “curvature” and “straightness” to describe moral concepts in daily life. For example, in the familiar Chinese word “是非曲直” (shì fēi qǔ zhí, which refers to the rights and wrongs), “curved and straight” maps to “wrong and right.” Specifically, “right” corresponds to “straight” and “wrong” corresponds to “curved,” in such a way that “straight” derives a positive meaning and “curved” derives a negative meaning. In real life, people often use words such as “正直” (zhèng zhí) and “刚直” (gāng zhí, which refers to upright and outspoken) to describe a person’s good qualities and words such as “扭曲” (nǐu qū, which refers to the distortion of personality) and “委曲求全” (wěi qū qíu qúan, which refers to stooping to compromise) to describe a person’s bad qualities. This shows that in Chinese culture, “straight” represents morality and “bent” represents immorality. In English, there are also etymological metaphors or lexical derivations that use the same metaphors for “straight” and “curved” as are used in the construction of Chinese characters. Through metaphorical derivation, the English word “straight” means “honest or upright,” and the word “slant” means “crooked or tilted,” giving the word “slant” the meaning of “crookedly reported” through metaphorical derivation [37]. Therefore, in this study, we aim to explore the following question: Is there a curvilinear metaphor for the processing of morality?

The present study explores the above questions on the basis of two experiments. Experiment 1 explored whether there is an unconscious automatic relationship between moral concepts and curvilinear graphs by using the implicit association test (IAT). The hypothesis of Experiment 1 was that, if there is a curvilinear metaphor for the representations of moral concepts, then participants should have significantly shorter reaction times when moral words are presented jointly with straight patterns and immoral words with curved patterns, and significantly longer reaction times when moral words are presented jointly with curved patterns and immoral words with straight patterns. Experiment 2 used the Stroop paradigm to further explore whether curved and straight physical features of the font affect the processing of moral concept words in Chinese. We used blackface characters with distinct linear features and Huaguang cursive characters with distinct curvilinear features to present moral and immoral words, respectively, aiming to provide further evidence to support the curvilinear metaphorical processing of moral concepts. Experiment 2 hypothesized that if the processing of moral concept words activates curvilinear metaphorical representation, then reaction times should be significantly shorter when moral words are presented in a font with a distinct linear sense (boldface) and immoral words are presented in a font with a distinct curvilinear sense (Huaguang cursive font); conversely, if moral words are presented in a font with a distinct curvilinear sense and immoral words are presented in a font with a distinct linear sense, then response times should be significantly longer.

## 2. Materials and Methods

### 2.1. Material Rating Experiment

#### 2.1.1. Word Ratings

##### Participants

Thirty-two college students, aged between 18 and 28 years (M = 21.63, SD = 2.57), were randomly selected to participate in the word-rating experiment, and none of them participated in the formal experiment. All participants were right-handed with normal or corrected-to-normal vision, without dyslexia, and were paid for their participation. The material ratings and following experiments were carried out in accordance with the recommendations of the Institute Ethics Committee, South China Normal University, with written informed consent obtained from all participants in accordance with the Declaration of Helsinki. The protocol was approved by the Institute Ethics Committee, South China Normal University.

##### Materials and Task

From the Dictionary of Modern Chinese adjectives, 40 four-character words concerning moral and immoral meanings were selected, among which 20 were moral words (e.g., 宁死不屈, nìng sǐ bù qū, which refers to choosing death before surrender) and 20 were immoral words (损人利己, sǔn rén lì jǐ, which refers to harming others to benefit oneself). Participants were instructed to rate these words on a 7-point scale in terms of (i) their familiarity with the words (1 = very unfamiliar; 7 = very familiar), (ii) their understanding of the words (1 = very little; 7 = a great deal), and (iii) the moral valence of the words (1 = very immoral; 7 = very moral).

#### 2.1.2. Pattern Ratings

##### Participants

A different group of 40 college students, aged between 17 and 25 years (M = 21.41, SD = 2.66), participated in the rating experiment regarding patterns. All participants were right-handed with normal or corrected-to-normal vision, without dyslexia, and were paid for their participation.

##### Materials and Task

There were 10 patterns with curved shapes and 10 patterns with straight shapes used. All patterns were generated using Adobe Photoshop CS6, and they were processed into 960 × 720 pixel resolution images with grey backgrounds. Participants were instructed to rate the complexity of the patterns on a 5-point scale (1 = very simple; 5 = very complex).

#### 2.1.3. Font Ratings

##### Participants

Thirty-five college students who had not participated in the previous experiments were randomly selected to participate in the font rating experiment. All participants were right-handed with normal or corrected-to-normal vision, without dyslexia, and were paid for their participation.

##### Materials and Task

Sixteen fonts (eight curved fonts and eight straight fonts) were selected from the font website. All fonts were presented through pseudowords. Participants were instructed to rate these words on two 7-point scales, one measuring the curved sense of the font and one measuring the straight sense of the font, with responses varying from 1 for very weak to 7 for very strong.

### 2.2. Experiment 1

In Experiment 1, the IAT paradigm was used to explore whether the mental representation of moral concepts is associated with curved or straight patterns. We predicted that reaction time would be faster when moral words were accompanied by a straight pattern and immoral words were accompanied by a curved pattern. than when moral words were accompanied by a curved pattern and immoral words were accompanied by a straight pattern.

#### 2.2.1. Participants

Thirty-six college students (20 females), who were right-handed, had normal or corrected-to-normal vision, without dyslexia, and aged between 18 and 22 years (M = 20.62, SD = 1.12) were randomly selected from South China Normal University to participate in Experiment 1, and they were paid for their participation.

#### 2.2.2. Materials

In accordance with the results of the material rating experiment, 20 words (10 moral and 10 immoral words) and 20 patterns (10 curved patterns and 10 straight patterns) were used as materials for Experiment 1.

#### 2.2.3. Design

Experiment 1 involved a 2 (types of word: moral words vs. immoral words) × 2 (types of pattern: curved pattern vs. straight pattern) factorial design. All variables were manipulated within subjects. The dependent variable was the mean reaction time in the compatible and incompatible trials.

#### 2.2.4. Procedure

E-prime 2.0 was used to present the IAT procedure, and all the experimental materials were processed into a grey background with 960 × 720 pixels. The specific experimental process was as follows. Step 1: A red fixation cross was presented in the center of the computer screen for 500 ms, then patterns were randomly presented in the center of the screen, and participants were asked to categorize the patterns via key press as either a straight pattern (“F” key) or a curved pattern (“J” key). Step 2: The words were presented randomly, and participants were instructed to classify the words as having either a moral or immoral meaning by using the same two key responses. Step 3: This step involved a compatible task practice in which the stimuli and tasks in Step 1 and Step 2 were combined. Participants were asked to classify the patterns as either curved or straight and then classify the words as either moral or immoral. Step 4: All stimuli and tasks in the compatible task testing in Step 4 were identical to those in Step 3. Step 5: All stimuli and tasks in step 5 were identical to those in Step 1, but the key press assignments were reversed. Step 6: In the incompatible task practice, all the stimuli and tasks were identical to those of Step 3, but the key-press assignments were reversed. Step 7: Incompatible task testing was conducted in which all the stimuli and tasks were identical to those in Step 6. The procedure is shown in Table 1.

### 2.3. Experiment 2

Experiment 2 aimed to explore whether the physical characteristics of fonts (curved or straight) affect the processing of moral or immoral words. We hypothesized that when moral words were presented in fonts with a strong sense of straightness and immoral words were presented in fonts with a strong sense of curvedness, the word classification reaction would be faster.

#### 2.3.1. Participants

Thirty-five college students (25 females), who were right-handed, had normal or corrected-to-normal vision, did not have dyslexia, and were aged between 18 and 22 years (M = 21.07, SD = 1.09) were randomly selected from South China Normal University, and they were paid for their participation.

#### 2.3.2. Materials

The word stimuli were identical to those in Experiment 1. In addition, 20 fonts (10 curved, 10 straight) were also selected on the basis of the material rating experiment.

#### 2.3.3. Design

Experiment 2 involved a 2 (types of font: curved font vs. straight font) × 2 (types of word: moral vs. immoral) factorial design. All variables were manipulated within subjects. The dependent variable was the reaction time and correct reaction rate of word categorization.

#### 2.3.4. Procedure

Experiment 2 adopted the Stroop paradigm, and was automatically presented by computer. All the experimental materials were processed into grey background pictures with 960 × 720 pixels. All words were presented in two fonts (one with a strong sense of straightness, such as见义勇为, jiàn yì yŏng wéi, which refers to ready to help others for a just cause, and one with a strong sense of curvedness, such as见利忘义, jiàn lì wàng yì, which refers to forgetting honour at the sight of money), at a size of 88 points and not bolded. The participants first needed to practice 20 trials, and then the computer provided feedback on whether their answers were correct. In the testing procedure, the computer recorded their accuracy and reaction time without giving feedback.

At the beginning of the procedure, a red fixation cross was displayed in the center of the computer screen for 500 ms, and then moral and immoral words were presented in random order and random fonts with either a strong sense of straightness or one of curvature. Participants were asked to categorize the words accurately and quickly by pressing the “F” key for moral words and pressing the “J” key for immoral words. Each testing word was presented for the maximum allowed reaction time of 2000 ms, followed by a 500 ms blank screen between each trial and then a red fixation cross (see Figure 1).

## 3. Results

### 3.1. Material Rating Experiment

#### 3.1.1. Word Ratings

Based on the ratings, 10 moral words with the highest moral valence and 10 immoral words with the lowest moral valence were selected as experimental materials. The results of the word ratings confirmed that there was a significant difference between the moral valence of moral words (6.64 ± 0.08) and that of immoral words (1.60 ± 0.13), t (9) = −64.71, *p* < 0.001. There was no significant difference between the two types of words regarding familiarity (6.10 ± 0.29; 6.01 ± 0.22), t (9) = 0.96, *p* = 0.362, nor between the understanding of moral words (6.12 ± 0.16) and that of immoral words (6.07 ± 0.22), t (9) = 0.39, *p* = 0.706.

#### 3.1.2. Pattern Ratings

The results showed that there was no significant difference in complexity between the two types of patterns (2.44 ± 0.65 and 2.19 ± 0.69 for curved and straight patterns, respectively), t (9) = 0.95, *p* = 0.367.

#### 3.1.3. Font Ratings

Based on the mean score of each font, we selected Huaguang cursive font as the font with the strongest sense of curvedness and Heiti font as the font with the strongest sense of straightness.

### 3.2. Experiment 1

Two participants were excluded due to error rates of higher than 20%. In addition, reaction times below 300 ms were recoded to 300 ms, while reaction times above 3000 ms were recoded to 3000 ms. A paired samples *t*-test was performed on the reaction times in both types of tasks.

The results showed that there was a significant difference in reaction times between compatible (674.42 ± 75.05) and incompatible (861.47 ± 137.58) conditions, t (1, 33) = −9.54, *p* < 0.001. The results suggest that there is a metaphorical connection between moral words and straight patterns and between immoral words and curved patterns. In Experiment 2, we tried to use the Stroop paradigm to provide further evidence for the use of curvilinear metaphors in moral concept.

### 3.3. Experiment 2

The data regarding correct reaction rates of lower than 80% and those for reaction times deviating by more than three standard deviations from the average were removed as outliers, resulting in the removal of 1.73% of the data. The mean reaction times in all conditions are shown in Figure 2.

The results regarding reaction times revealed no significant main effect of the type of font, F (1, 34) = 0.18, *p* = 0.676, η^2^ = 0.01. The main effect of word types was significant, F (1, 34) = 21.97, *p* < 0.001, η^2^ = 0.39. We also found a significant interaction between types of font and types of word, F (1, 34) = 6.93, *p* < 0.050, η^2^ = 0.17, which revealed that the reaction time (723.27 ± 102.07) for moral words presented in a straight font was significantly shorter than that for moral words presented in a curved font (744.30 ± 110.62), F (1, 34) = 4.69, *p* < 0.050, η^2^ = 0.12. However, the reaction time to immoral words presented in a curved font (790.30 ± 137.83) showed no significant difference from that presented in a straight font (804.81 ± 145.67), F (1, 34) = 1.82, *p* = 0.187, η^2^ = 0.05. When using a straight font, the reaction time to moral words (723.27 ± 102.07) was significantly shorter than that to immoral words (804.81 ± 145.67), F (1, 34) = 25.99, *p* < 0.050, η^2^ = 0.43. When using a curved font, the reaction time to moral words (744.30 ± 110.62) was also significantly shorter than that to immoral words (790.30 ± 137.83), F (1, 34) = 10.29, *p* < 0.050, η^2^ = 0.23.

The results of Experiment 2 partially support the initial experimental hypothesis, demonstrating that the mental representation of moral concepts is associated with straight and curved patterns in Chinese culture.

## 4. Discussion

Previous studies on moral metaphors focused on the dimensions of cleanliness, color, light and darkness, and space, with fewer empirical studies focusing on curvilinear metaphors. This study used the IAT (Experiment 1) and the Stroop paradigm (Experiment 2) to explore whether there are curvilinear metaphors employed in the processing of moral concepts under the Chinese cultural context. Consistent with the expectations and hypotheses, the reaction times for the IAT in Experiment 1 were significantly shorter when moral words were accompanied by straight patterns and when immoral words were accompanied by curved patterns, and they were significantly longer when moral words were accompanied by curved patterns and immoral words were accompanied by straight patterns. The results of the study showed that people associated “straight” with moral and “curved” with immoral in implicit unconscious processing and were therefore more likely to associate moral words when they saw straight lines and immoral words when they saw curved lines. This is the same principle as that in people’s perception of Chinese words such as “是非曲直” (shì fēi qǔ zhí, which refers to rights and wrongs).

The results of this study are similar to those of existing metaphor studies. Some researchers collected English language and image evidence, and through semantic material analysis found that five pairs of moral space metaphors exist in both English and Chinese cultural contexts, which include the curved and straight spatial metaphors [38]. Some researchers studied the metaphorical connection between geometric shapes (square and circle) and personality traits (integrity and cunning) and found that in personality classification tasks, memory tasks, and personality judgement tasks, participants were more inclined to associate squares with integrity, while roundness was often linked with guilt [39]. At the same time, some researchers found that people have a greater preference for straight lines after recalling moral events than they do after recalling immoral events [40]. Similar results were found in a semantic recognition task, where participants were asked to judge the moral properties contained in standard or distorted Chinese characters using the Stroop paradigm, and it was found that when presented with standard Chinese characters, participants were able to identify morally significant characters faster and more accurately, while when they were presented with distorted Chinese characters, they were able to identify characters with immoral meanings faster and more accurately [41].

In addition, this tendency to associate linear objects with positive concepts and curved objects with negative concepts can also be explained from the perspective of evolutionary psychology. To adapt to specific environments and successfully breed, early humans gradually evolved from crawling to walking upright [42]. The evolutionary process is mainly manifested through the transformation of the human body from a bent state to a straight state, and upright walking gradually helped our human ancestors transition from obscurity to civilization, and morality, art and wisdom developed accordingly. This is likely the reason that we first developed a fondness for straightness because an upright body represents civilization and advancement, and a bent body represents obscurity and backwardness.

Experiment 2 further used the Stroop paradigm to conduct word sense judgements on the basis of moral or immoral words as presented in either straight or curved feature fonts. It was found that the processing of moral words was significantly faster when those words were presented in a linear font than in a curved font, and the results suggest that linear fonts activate moral concepts and thus facilitate the word-meaning judgements of moral words. This phenomenon can also be explained on the basis of the concept of “cognitive processing fluency”, proposed by Fazio in 2001, which refers to the fact that people process two things with a consistent mental meaning more easily and with faster reaction times, and respond more slowly to two things with inconsistent mental meanings [43]. Fonts with a strong sense of straight lines have a psychological impact that is similar to that of moral concepts; therefore, when moral words are presented in conjunction with fonts with a strong sense of straight lines, people process them more smoothly and with faster reaction times. However, the curved and straight font effect exists only for moral words and disappears for immoral words. This may be because orthographic characters are more aligned with people’s reading habits. In our daily life, we have more contact with orthographic characters, and most such characters appear in textbooks and other formal books as a kind of standard or norm. Therefore, these characters are more likely to have a metaphorical connection with moral concepts. However, the metaphorical effect of curves and straightness disappears at the level of immoral words, probably because fonts with a strong sense of curves do not conform to people’s conventional understanding of curves in life. Most of the curves we come into contact with in our lives are irregular and inaesthetic. Distorting items makes it easier to cause moral discomfort, and fonts with a strong sense of curvature are generally more standardized, so it may be more difficult to activate people’s associations with immoral concepts.

In summary, on the basis of previous studies, this study explores moral metaphors in the Chinese context from multiple perspectives through implicit association experiments and the Stroop paradigm, and the results verify that moral metaphors exist in the Chinese context. The results showed that moral concepts and physical shapes are interlinked at the level of psychological representation. Through metaphor mapping, the use of straightness can deepen the understanding of abstract moral concepts, and the use of curvature can deepen the understanding of abstract immoral concepts. From a theoretical point of view, the results of this study help to broaden the field and to open up a new direction in the study of metaphors for moral concepts. At the same time, the current work also has a certain practical significance. Metaphor is pervasive in everyday communication, and is known to help people understand complex topics and communicate efficiently [44]. A concrete concept of shape can deepen the understanding of abstract concepts of moral character [30]; hence, in moral education for primary school students, we can use straight lines and shapes to strengthen the learning of moral behavior, and use curved patterns to strengthen knowledge of unethical behaviors. Similarly, we can also use straight-line backgrounds in public facilities and advertisements to increase people’s sense of association with moral behaviors, and use curved patterns and backgrounds to enhance their sense of prohibition of unethical behaviors. In addition, the straight metaphor for morality occurs in implicit automatic processing and an automatic connection with the unconscious. People always subconsciously regard “curvy” as “immoral” and “straight” as “moral”. There is a stereotype in people’s moral cognition that suggests people hold a view of handwriting as representing the person writing, and thus “distorted handwriting” becomes associated with “misconduct”. This phenomenon was also found in Chinese physiognomy, which showed that it is easier to view a person with a straight nose as virtuous and a person with a curved nose as immoral. The reason people are generally unaware of the prevalence of metaphors in language and their lives is because metaphors are by their very nature hidden since, unlike similes, they do not employ special linguistic markers [45]. If the existence of curvilinear metaphors for moral concepts is demonstrated in this study, it can help people to understand more about moral metaphors, so that we can intentionally control these metaphorical elements and sound the alarm in order to reduce stereotypes and other biased behaviors in real life.

Both the IAT and the Stroop Test used in this study tentatively confirmed the existence of curvilinear metaphors in the representation of moral concepts, but they were not sufficiently comprehensive and in-depth, leaving many issues that deserve to be further explored and discussed. First, in this research on the metaphorical representation of moral concepts, moral words and immoral words were selected as experimental materials. Future research should try to use pictures describing moral and immoral behaviors or play videos depicting moral and immoral behaviors to replace the traditional presentation of word stimuli to further explore the scope of straight metaphors in the representation of moral concepts. Second, this study was conducted against the background of Chinese culture, and the subjects were limited to college students. Future research should explore the cross-cultural validity and cross-age validity of the findings. Finally, this study only uses behavioral experiments to show that there are metaphorical representations of moral concepts, and does not determine the underlying mechanism. Follow-up research can further explore this aspect.

## 5. Conclusions

The present findings indicate that the mental representation of moral concepts is associated with straight and curved patterns in Chinese culture, in which Chinese participants tend to associate moral words with straight patterns and fonts and immoral words with curved patterns and fonts.

## Figures and Tables

**Figure 1 behavsci-13-00295-f001:**
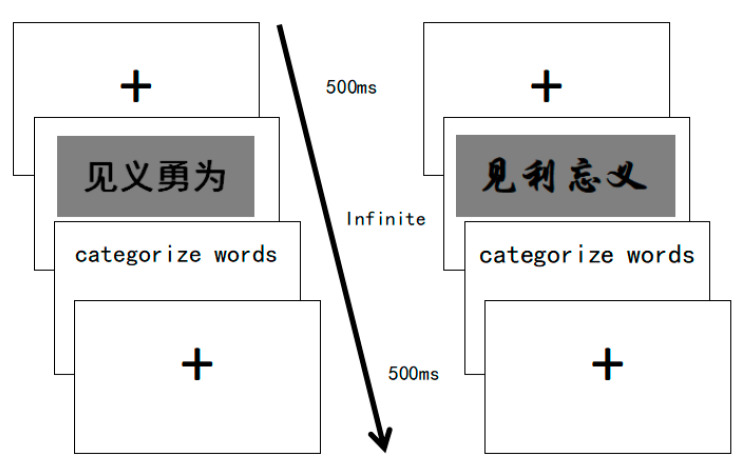
The experimental procedure for Experiment 2.

**Figure 2 behavsci-13-00295-f002:**
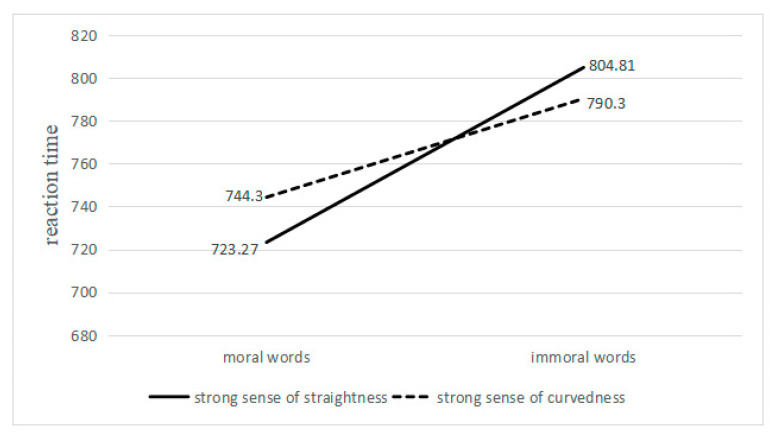
Interaction diagram of word type and font type.

**Table 1 behavsci-13-00295-t001:** The IAT test procedures.

Step	No. of Trials	Practice/Test	F Key Response	J Key Response
1	20	Practice	Straight patterns	Curved patterns
2	20	Practice	Moral words	Immoral words
3	20	Practice	Straight patterns + moral words	Curved patterns + immoral words
4	40	Test	Straight patterns + moral words	Curved patterns + immoral words
5	20	Practice	Curved patterns	Straight patterns
6	20	Practice	Curved patterns + moral words	Straight patterns + immoral words
7	40	Test	Curved patterns + moral words	Straight patterns + immoral words

## Data Availability

Data from the study are available upon request.

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
