# Peer review of "Keeping Morality “on the Straight” and Never “on the Bend”: Metaphorical Representations of Moral Concepts in Straightness and Curvature"

_behavsci, 2023, doi:10.3390/bs13040295_

Round 1

Reviewer 1 Report

This paper talks about the possibility of studying the moral metaphor in relation to the concept of “straight” and “curve”, using the Implicit Association Test and the Test of Stroop.

I accept with minor revisions.

The paper is well done and quite complete. However, in the introduction there are other important studies related to the relation between morality and compulsive behavior in healthy subjects that could be cited. As well as there are important studies with non-invasive brain stimulation techniques that suggest an important link between the moral, social and core disgust. Please pay more attention to this literature.

A critical point of the paper concerns the organization. There are too many sub-paragraphs which tend to confuse the reader.

I would like to suggest to the author to collect the following paragraphs into just two paragraphs:

2.1.1.Word Ratings

2.1.1.1.Participants

2.1.1.2.Materials and task,

2.1.2.1.Participants

2.1.2.2.Materials and task.

I suggest writing them like this:

2.1 Material Rating Experiment

2.1.1 Word Ratings (Include 2.1.1.1.Participants 2.1.1.2.Materials and task)

2.2.2 Pattern Ratings (Include 2.1.2.1.Participants 2.1.2.2.Materials and task)

Finally, a limitation of this study is related to the absence of an English translation of the Chinese word. In the introduction and also in the other paragraphs of the paper there are many Chinese words that lack a specific translation.

Here some examples:

Pag. 2 “In daily life, people often use words such as “清身洁己” (qīng shēn jíe jǐ) and

冰清玉” (bīng qīng yù jíe) to describe people who behave morally and words such as “脏心烂肺” (zāng xīn làn fèi) and “脏官污吏” (zāng gūan wū lì) to describe people who behave immorally”.

Pag. 3 “Another metaphorical relationship similar to the moral-colour metaphor is the moral-lucidity metaphor [15;16] . Chinese people often use expressions such as “弃暗投明” (qì àn tóu míng), “明人不做暗事” (míng rén bú zùo àn shì) and “柳暗花明” (lǐu àn hūa míng), implying that “brightness” contains a positive meaning and “dark” 116 contains a derogatory meaning”.

Pag. 4 “. In real life, people often use words such as “正直” (zhèng zhí) and “刚直” (gāng zhí) to describe a person’s good qualities and words such as “扭曲” (nǐu qū) and “委曲求全” (wěi qū qíu qúan) to describe a person’s bad qualities”.

Pag. 5 “From the Dictionary of Modern Chinese adjectives, 40 four-character words concerning moral and immoral meanings were selected, among which 20 were moral words (e.g., 宁死不屈, nìng sǐ bù qū) and 20 were immoral words (损人利己, sǔn rén lì jǐ).”

I think that without a specific translation of the stimuli, it is difficult to appreciate the research.  Please provide an English translation of the Chinese words, and if it is possible, please explain all the words (moral and immoral) you used as stimuli.

Author Response

Dear Editor and Reviewers,

Thanks a lot for your letter and for the reviewers’ comments concerning our manuscript entitled “Keeping morality ‘on the straight’ and never ‘on the bend’: Metaphorical representations of moral concepts in straightness and curvature” (behavsci-2260403). The comments are very valuable and helpful for further revising our paper. We have studied the comments carefully and have made substantial revisions according to the comments. Revised portions were highlighted and marked in blue in the paper. The main revisions in the paper and the responds to the reviewers’ comments are as following.

Thanks again for the insightful comments.

Sincerely,

The reply to the comments of Reviewer 1:

Comment 1:

The paper is well done and quite complete. However, in the introduction there are other important studies related to the relation between morality and compulsive behavior in healthy subjects that could be cited. As well as there are important studies with non-invasive brain stimulation techniques that suggest an important link between the moral, social and core disgust. Please pay more attention to this literature.

Responds:

Thanks for the comments. Following your suggestion, we further found several literatures related to moral metaphor, moral disgust and core disgust which have been cited in the introduction. The revisions and supplementary literatures were listed as follow:

Previous study also applied eye-tracking technology and tracked pupil size before and after hand washing as an objective measurement for subjects’ emotional arousal, and they found that compare to washing hand group, no washing group had extreme rating scores for morality. Meanwhile, the eye-tracking results also showed a decreased data pattern of pupil diameter after hand washing [11]. In the separated Stroop paradigm and the event-related potentials (ERPs) experiments, participants recognized moral words faster after reading environmentally clean sentences than neutral sentences, and the amplitude was more positive in the P300 and N400 time windows. Meanwhile, they also estimated moral words faster after reading self-clean sentences than neutral sentences, and the amplitude was more positive in the P300 and N400 time windows [12; 13]

In another research, the cognitive mechanisms underlying moral metaphors on the vertical dimension was explored, and the results revealed a stronger association of “moral-up, immoral-down” when the emotionality of moral and immoral words was high than those in low emotionality conditions [26].

The bidirectional representational relationship between moral concepts and physical attributes shows that core disgust, such as dirty and stink, can affect behavior and spatial dimensions of moral judgment. For example, research showed that both moral violations involve and do not involve impurity can promoted the detection of disgusted faces [34], and moral violation with non-purity cues elicited higher severity of moral judgments than those without non-purity cues [35]. However, this relation-ship was modulated by the degree of core disgust priming, which one research had found that moral severity ratings for moral transgressions after the presentation of neural images were significantly higher those after the presentation of highly disgust images [36].

Holyoak, K.J.; Dušan Stamenkovi. Metaphor comprehension: A critical review of theories and evidence. Psychol. Bull. 2018, 144, 6, 641-671.https://doi.org/10.1037/bul0000145.

Kaspar, K.; Krapp, V.; König, P. Hand washing induces a clean slate effect in moral judgments: A pupillometry and eye-tracking study.Sci Rep . 2015, 5, 10-17. https://doi.org/10.1038/srep10471.

Ding, F.Q.; Sun, Y.S.; Wang, X.F. The metaphor of self/environmental cleanliness in the case of moral concepts: an event-related potential study. J. Cogn. Psychol. 2023, 35, 1, 36-46. https://doi.org/10.1080/20445911.2022.2104859.

Ding, F.Q.; Tian, X.Y.; Wang, X.M.; Liu, Z. The consistency effects of the clean metaphor of moral concept and dirty metaphor of immoral concept.J. Psychophysiol. 2020, 34, 4, 214-223. https://doi.org/10.1027/0269-8803/a000249.

Zhai, D.X.; Guo, Y.L.; Lu, Z.Y. A dual mechanism of cognition and emotion in processing moral-vertical metaphors. Front. Psychol. 2018, 9, 1554. doi: 10.3389/fpsyg.2018.01554.

Olatunji, B.O.; Puncochar, B.D. Effects of disgust priming and disgust sensitivity on moral judgement. Int. J. Psychol. 2016, 51, 2, 102-108.

Jiang, S.Y.; Peng, M.; Wang, X.H. Different influences of moral violation with and without physical impurity on face processing: An event-related potentials study. Plos One. 2020, 1-17. https://doi.org/10.1371/journal.pone.

Tao, D.; Leng, Y.; Huo, J.M.; Peng, S.H.; Xu, J.; Deng, H.H. Effects of core disgust and moral disgust on moral judgment: An event-related potential study. Front. Psychol. 2022, 13, 1-12.

Thibodeau, P.H.; Matlock, T.; Flusberg, S.J. The role of metaphor in communication and thought. Lang. Linguist. Compass. 2019, 13, 5, 10-20. https://doi.org/10.1111/lnc3.12327.

Gandolfo, S. Metaphors of metaphors: Reflections on the use of conceptual metaphor theory in premodern Chinese texts. Metaphors of Metaphors. 2019, 18, 323-345. https://doi.org/10.1007/s11712-019-09669-0.

Comment 2:

A critical point of the paper concerns the organization. There are too many sub-paragraphs which tend to confuse the reader.

I would like to suggest to the author to collect the following paragraphs into just two paragraphs:

2.1.1.Word Ratings

2.1.1.1.Participants

2.1.1.2.Materials and task,

2.1.2.1.Participants

2.1.2.2.Materials and task.

I suggest writing them like this:

2.1 Material Rating Experiment

2.1.1 Word Ratings (Include 2.1.1.1.Participants 2.1.1.2.Materials and task)

2.2.2 Pattern Ratings (Include 2.1.2.1.Participants 2.1.2.2.Materials and task)

Responds:

Thanks for the suggestions. In order to make the structure of the article clearer, we have collect the sub-paragraphs into just one big paragraphs.

Comment 3:

Finally, a limitation of this study is related to the absence of an English translation of the Chinese word. In the introduction and also in the other paragraphs of the paper there are many Chinese words that lack a specific translation.

Here some examples:

Pag. 2 “In daily life, people often use words such as “清身洁己” (qīng shēn jíe jǐ) and

“冰清玉洁” (bīng qīng yù jíe) to describe people who behave morally and words such as “脏心烂肺” (zāng xīn làn fèi) and “脏官污吏” (zāng gūan wū lì) to describe people who behave immorally”.

Pag. 3 “Another metaphorical relationship similar to the moral-colour metaphor is the moral-lucidity metaphor [15;16] . Chinese people often use expressions such as “弃暗投明” (qì àn tóu míng), “明人不做暗事” (míng rén bú zùo àn shì) and “柳暗花明” (lǐu àn hūa míng), implying that “brightness” contains a positive meaning and “dark” 116 contains a derogatory meaning”.

Pag. 4 “. In real life, people often use words such as “正直” (zhèng zhí) and “刚直” (gāng zhí) to describe a person’s good qualities and words such as “扭曲” (nǐu qū) and “委曲求全” (wěi qū qíu qúan) to describe a person’s bad qualities”.

Pag. 5 “From the Dictionary of Modern Chinese adjectives, 40 four-character words concerning moral and immoral meanings were selected, among which 20 were moral words (e.g., 宁死不屈, nìng sǐ bù qū) and 20 were immoral words (损人利己, sǔn rén lì jǐ).”

I think that without a specific translation of the stimuli, it is difficult to appreciate the research.  Please provide an English translation of the Chinese words, and if it is possible, please explain all the words (moral and immoral) you used as stimuli.

Responds:

Thanks for the suggestions. We have provided the specific translations of the Chinese words. The revisions could be summarized as follow:

Pag. 2 In daily life, people often use words such as “清身洁己” (qīng shēn jíe jǐ, refers to maintaining one's own moral integrity and physical conduct ) and “冰清玉洁” (bīng qīng yù jíe, refers to noble and pure personality traits) to describe people who behave morally and words such as a脏心烂肺” (zāng xīn làn fèi, refers to people who are sordid and devious) and “脏官污吏” (zāng gūan wū lì, refers to corrupt officials).

Pag. 3 Chinese people often use expressions such as “弃暗投明” (qì àn tóu míng, refers to forsake darkness for light) and “明人不做暗事” (míng rén bú zùo àn shì, refers to an honest man does not do anything underhand).

Pag. 4 For example, in the familiar Chinese word “是非曲直” (shì fēi qǔ zhí, refers to the rights and wrongs), “curved and straight” maps to “wrong and right.” Specifically, “right” corresponds to “straight” and “wrong” corresponds to “curved,” in such a way that “straight” derives a positive meaning and “curved” derives a negative meaning. In real life, people often use words such as “正直” (zhèng zhí) and “刚直” (gāng zhí, refers to upright and outspoken) to describe a person’s good qualities and words such as “扭曲” (nǐu qū, refers to the distortion of personality) and “委曲求全” (wěi qū qíu qúan, refers to stoop to compromise) to describe a person’s bad qualities.

Pag. 5 40 four-character words concerning moral and immoral meanings were selected, among which 20 were moral words (e.g., 宁死不屈, nìng sǐ bù qū, refers to choose death before surrender) and 20 were immoral words (损人利己, sǔn rén lì jǐ, refers to harm others to benefit oneself).

Once again, thanks a lot for your useful comments and suggestions.

Reviewer 2 Report

Three key pieces are important:

1. Need to set up some comparative terms outside the Chinese language.  In other words, what are obvious or possible synonyms in other cultures?

2.  Improve your bibliography.  I realize you are adding to the literature, but you build upon old sources.  There have to be a cache of references to your topic that are more contemporary.

3.  So what?  What is the intent of your study?  Is it merely to share some insights with the global village from a particular culture.  It may be in your interest and that of the journal's readers to see some application to moral quandaries.  

Thank you! 

Author Response

Dear Editor and Reviewers,

Thanks a lot for your letter and for the reviewers’ comments concerning our manuscript entitled “Keeping morality ‘on the straight’ and never ‘on the bend’: Metaphorical representations of moral concepts in straightness and curvature” (behavsci-2260403). The comments are very valuable and helpful for further revising our paper. We have studied the comments carefully and have made substantial revisions according to the comments. Revised portions were highlighted and marked in blue in the paper. The main revisions in the paper and the responds to the reviewers’ comments are as following.

Thanks again for the insightful comments.

Sincerely,

The reply to the comments of Reviewer 2:

Comment 1:

Need to set up some comparative terms outside the Chinese language.  In other words, what are obvious or possible synonyms in other cultures?

Responds:

Thanks for the suggestions. We have search some synonyms and phrases in English. The revisions could be summarized as follow.

In daily life, people often use linguistic metaphors to express the moral concepts in straightness and curvature. For example, “courage is the ladder on which all the other virtues mount” and “by hook or by crook”.

Comment 2:

Improve your bibliography. I realize you are adding to the literature, but you build upon old sources. There have to be a cache of references to your topic that are more contemporary.

Responds:

Thanks for the suggestions. We have search a series of literatures in the last 4 years, and these latest literatures have also been cited in the manuscript, and listed as follow.

Ding, F.Q.; Sun, Y.S.; Wang, X.F. The metaphor of self/environmental cleanliness in the case of moral concepts: an event-related potential study. J. Cogn. Psychol. 2023, 35, 1, 36-46. https://doi.org/10.1080/20445911.2022.2104859.

Ding, F.Q.; Tian, X.Y.; Wang, X.M.; Liu, Z. The consistency effects of the clean metaphor of moral concept and dirty metaphor of immoral concept.J. Psychophysiol. 2020, 34, 4, 214-223. https://doi.org/10.1027/0269-8803/a000249.

Jiang, S.Y.; Peng, M.; Wang, X.H. Different influences of moral violation with and without physical impurity on face processing: An event-related potentials study. Plos One. 2020, 1-17. https://doi.org/10.1371/journal.pone.

Tao, D.; Leng, Y.; Huo, J.M.; Peng, S.H.; Xu, J.; Deng, H.H. Effects of core disgust and moral disgust on moral judgment: An event-related potential study. Front. Psychol. 2022, 13, 1-12.

44.Thibodeau, P.H.; Matlock, T.; Flusberg, S.J. The role of metaphor in communication and thought. Lang. Linguist. Compass. 2019, 13, 5, 10-20. https://doi.org/10.1111/lnc3.12327.

45.Gandolfo, S. Metaphors of metaphors: Reflections on the use of conceptual metaphor theory in premodern Chinese texts. Metaphors of Metaphors. 2019, 18, 323-345. https://doi.org/10.1007/s11712-019-09669-0.

Comment 3:

So what? What is the intent of your study? Is it merely to share some insights with the global village from a particular culture. It may be in your interest and that of the journal's readers to see some application to moral quandaries.

Responds:

Thanks for the suggestions. We have made some revisions in the discussion section of the manuscript to detail the application of our research in daily life.

In the moral education for primary school students, we can use straight lines and shapes to strengthen the moral behavior learning, and use curved patterns to strengthen the knowledge of unethical behaviors. Similarly, we can also use straight-line backgrounds in the public facilities and advertisements to increase people’s sense of association with moral behaviors, and use curved patterns and back-grounds to enhance the prohibition of unethical behaviors.

In addition, the straight metaphor for morality occurs in implicit automatic processing and an automatic connection with the unconscious. People always subconsciously regard “curvy” as “immoral” and “straight” as “moral”. There is a stereotype in people’s moral cognition that suggest people hold a view of handwriting representing the person writing, and thus “distorted handwriting” becomes associated with “misconduct”. This phenomenon was also found in Chinese physiognomy, which showed it is easier to view a person with a straight nose as virtuous and a person with a curved nose as immoral. The reason people are generally unaware of the prevalence of metaphors in language and their lives is because metaphors are by their very nature hidden since, unlike similes, they do not employ special linguistic markers. If the existence of curvilinear metaphors for moral concepts is demonstrated in this study, it can help people to understand more about moral metaphor, then we can intentionally control these metaphorical elements and sound the alarm, then reduce stereotypes and other biased behaviors in real life.

Once again, thanks a lot for your useful comments and suggestions.

Round 2

Reviewer 2 Report

Excellent revision!  Thanks for being so attentive to my concerns.  I believe you will make an excellent contribution to the field!